# Control of epithelial tissue organization by mRNA localization

**Devon E. Mason** [1] ✉, **Thomas D. Madsen** [1], **Alexander N. Gasparski** [1],
**Dong Kong** [2], **Neal Jiwnani**[1], **Terry Lechler**[3,4], **Jadranka Loncarek**[2],
**Roberto Weigert** [1], **Ramiro Iglesias-Bartolome** [1] **& Stavroula Mili** [1] ✉

mRNA localization to specific subcellular regions is common in mammalian cells but poorly understood in terms of its physiological roles. This study demonstrates the functional importance of *Net1* mRNA, which we find prominently localized at the dermal-epidermal junction (DEJ) in stratified squamous epithelia. *Net1* mRNA accumulates at DEJ protrusion-like structures that interact with the basement membrane and connect to a mechanosensitive network of microfibrils. Disrupting *Net1* mRNA localization in mouse epithelium alters DEJ morphology and keratinocyte-matrix connections, affecting tissue homeostasis. mRNA localization dictates the cortical accumulation of the Net1 protein and its function as a RhoA GTPase exchange factor (GEF). Altered RhoA activity is in turn sufficient to alter the ultrastructure of the DEJ. This study provides a high-resolution in vivo view of mRNA targeting in a physiological context. It further demonstrates how the subcellular localization of a single mRNA can significantly influence mammalian epithelial tissue organization, thus revealing an unappreciated level of post-transcriptional regulation that controls tissue physiology.

mRNA localization has emerged as a prevalent level of post-transcriptional regulation not only in model systems but also in higher organisms. Indeed, in diverse mammalian cell types, a large fraction of the transcriptome is targeted to specific subcellular destinations and adopts distinct distribution patterns[1–8]. Nevertheless, whether the localization of individual mRNAs is important for mammalian physiology remains poorly characterized, especially in tissues outside the nervous system[9,10].

A well-studied localization pathway targets mRNAs to peripheral protrusions of mammalian mesenchymal cells through active kinesin-dependent trafficking on microtubules[1,11–13]. Localization to protrusions of mRNAs, such as *NET1* and *RAB13*, plays an important role in mesenchymal migration in cultured cell systems[14–17]. Mechanistically, even though proteins like NET1 and RAB13 do not remain concentrated close to their translation site, the location of their mRNAs influences the eventual destination and function of the encoded proteins by specifying where in cytoplasm the protein is translated. Protein synthesis within a particular local micro-environment promotes specific co-translational interactions of the nascent proteins thus guiding binding partner selection and eventual protein function[14,16,18]. In the case of NET1, localization of *NET1* mRNA at protrusions promotes NET1 protein association with a membrane-bound scaffold[16]. Plasma membrane-associated NET1 acts as a guanine nucleotide exchange factor (GEF) for the small GTPase RhoA, a central regulator of the cytoskeleton[19,20]. In contrast, perinuclear *NET1* mRNA promotes nascent NET1 binding to importins leading to its nuclear sequestration. In this way, the location of the *NET1* mRNA modulates the nucleo-cytoplasmic distribution of NET1 protein and consequently influences cytoskeletal dynamics and complex cellular processes like mesenchymal cell migration in vitro[16]. Protrusion-localized mRNAs, including *NET1*, have also been observed at the basal surface of in vitro cultured epithelial cells as well as in intestinal enterocytes in vivo[2,3,8]. However,

[1]Laboratory of Cellular and Molecular Biology, Center for Cancer Research, National Cancer, Institute, NIH, Bethesda, MD, USA. [2]Cancer Innovation Laboratory, Center for Cancer Research, National Cancer institute, NIH, Frederick, MD, USA. [3]Department of Dermatology, Duke University Medical Center, Durham, NC, USA. [4]Department of Cell Biology, Duke University Medical Center, Durham, NC, USA. ✉e-mail: devon.mason@nih.gov; voula.mili@nih.gov

the functional significance of this epithelial basal localization is unknown.

Here, we examine the functional role of protrusion mRNA localization in mouse epithelial physiology. We find that, strikingly, protrusion-localized mRNAs are targeted to poorly characterized protrusion-like structures formed by basal keratinocytes in mammalian stratified epithelial tissue. These formations at the dermal-epidermal junction (DEJ) are connected to mechanosensitive microfibrils in the extracellular matrix. Using specific, sequence-blocking anti-sense oligos to prevent *Net1* mRNA localization to the DEJ, we show that *Net1* mRNA location is necessary for maintaining DEJ architecture and epithelial homeostasis through the RhoA pathway. This study connects mRNA localization to unexplored aspects of epithelial physiology and highlights this prevalent level of post-transcriptional regulation as an important contributor to tissue function in higher organisms.

## Results

### Net1 mRNA localizes to the basal cell membrane in epithelia

To probe the function of protrusion-localized mRNAs in epithelia in vivo, we first surveyed the subcellular localization of several protrusion-localized mRNAs[12,21] (i.e., *Net1, Cyb5r3, Palld, Kif1c, Pkp4*) across various mouse epithelial tissues that are architecturally distinct. These tissues included monolayered tubular epithelial structures (intestine and kidney) as well as stratified squamous epithelia (tongue and skin) (Fig. 1a). Robust signals were detected with transcript-specific probes, but not with a negative control probe against a bacterial mRNA (DapB) (Fig. 1a and Supplementary Fig. 1). To allow spatial quantitative assessments, RNA detection was accompanied by staining with a combination of wheat germ agglutin (WGA) and β-catenin to delineate overall tissue architecture and cell borders (Fig. 1b and Supplementary Fig. 2a, b). To permit comparison across tissues, we focused in each case on the basal layer of cells in contact with the extracellular basement membrane (BM). The BM was demarcated by WGA providing a comparable basal boundary between BM-associated cells in different tissues (Fig. 1a, b; BM marked by a yellow dashed line). We manually segmented the BM-associated cell layer (Fig. 1c and Supplementary Fig. 2a, b; segmented basal cell layer marked by a red dashed line) and RNA signal was quantified along the apico-basal axis (Fig. 1d), providing a measure of subcellular mRNA distribution within a layer of similarly polarized cells. We define the RNA amount in the bottom 30% of this layer as the 'basal mRNA fraction' (Fig. 1c, d and Supplementary Fig. 2a). Interestingly, most of the tested mRNAs accumulate in the basal part of the BM-attached cell layer in all tissues tested, whereas a non-targeted control transcript, *Gapdh* mRNA, was comparatively diffuse (Fig. 1e and Supplementary Fig. 2c). There was also evident variability in basal mRNA targeting both among tissues and between protrusion-localized mRNAs, suggesting transcript- and tissue-specific differences in the magnitude of mRNA localization. Nevertheless, mRNAs like *Net1* and *Cyb5r3* emerged as consistently basally localized mRNAs in all tissues tested.

To explore the relevance of these candidates to tissue physiology, we surveyed published RNA-seq and tissue RNA imaging datasets. Interestingly, *Net1* is specifically enriched in basal keratinocytes of the skin and tongue; *Net1* is also nearly 4-times as abundant as the 5 next most highly expressed Rho-family GEFs (Supplementary Fig. 2d, e)[22–24]. Furthermore, *Net1* expression is 9-times higher in keratinocytes than dermal fibroblasts of the skin pointing to a potential important function in the epidermal compartment (Supplementary Fig. 2d)[22]. In addition, we characterized the in vivo *NET1* mRNA distribution in stratified epithelial tissues of other species by examining primate tongue and human skin biopsies. Importantly, we again found in these cases that *NET1* mRNA is significantly enriched near the basal cell membrane of BM-attached keratinocytes suggesting that basal *NET1* mRNA localization is evolutionarily conserved (Fig. 1a, e). We thus

focused on the mouse *Net1* mRNA for further molecular and functional characterization utilizing the stratified tongue epithelium as an amenable experimental system.

The stratified squamous epithelium of the tongue consists of epithelial cells at different proliferation and differentiation states. Basal keratinocytes attached to the BM can enter the cell cycle and upregulate proliferating cell nuclear antigen (PCNA) during S/G2/M-phases[25]. To see whether *Net1* mRNA localization differs between quiescent and proliferating basal cells, we measured basal *Net1* accumulation in PCNA^Hi and PCNA^Lo cells (Fig. 1f). *Net1* distribution was not significantly different between cell proliferation states, based on this cell division marker (Fig. 1g). Basal Krt5+ cells can additionally transition to the suprabasal layer while upregulating differentiation markers like keratin 13 (Krt13) in the mouse tongue. As a result, such Krt13+ cells eventually lose contact with the BM. Quantification of total *Net1* mRNA amount (by RNA-FISH) in segmented epithelial layers showed that *Net1* mRNA amount is highest in the BM-attached basal layer and downregulated in suprabasal differentiated keratinocytes (Supplementary Fig. 2e). Interestingly, we could detect cells that appear to be in the process of delaminating, since they exhibit basally-oriented Krt13 cellular extensions intercalating between Krt5+ cells (Fig. 1h). Such cells exhibited accumulation of *Net1* mRNA at the tips of these extensions, albeit infrequently (Fig. 1h inset). We interpret this to suggest that *Net1* mRNA localization close to the basal plasma membrane is transiently maintained in delaminating keratinocytes concomitantly with an overall *Net1* downregulation during differentiation. Overall, these data suggest that both *Net1* mRNA expression and subcellular localization is notably elevated in BM-contacting keratinocytes, regardless of their proliferation state. We therefore focused our attention on characterizing *Net1* mRNA and its function in basal keratinocytes.

### Net1 mRNA accumulates in microfiber anchored protrusions

Basal cells of the tongue reside at the interface between the oral epithelium and the underlying connective tissue. This interface in stratified epithelia has been primarily studied in the skin and referred to as the dermal-epidermal junction (DEJ). We will be using the same terminology here, given the parallels between oral and skin tissue[26]. The DEJ includes the basal plasma membrane and specialized adhesions called hemidesmosomes, which connect the epithelium to the BM and the dermal extracellular matrix (ECM) through fibrillar anchoring complexes[27,28]. The DEJ has important roles in maintaining the epithelium both by acting as a reservoir for signaling molecules and by providing structural support[29,30].

We performed high-resolution confocal imaging of the DEJ, because of the pronounced basal accumulation of the *Net1* mRNA. We specifically visualized the basal cell membrane through staining with integrin a6 (Itga6), a core hemidesmosome component. Strikingly, high-resolution imaging revealed a high degree of topographical variability of the basal membrane which consisted of numerous micron- or submicron-wide protrusion-like structures that interdigitated with the BM (Fig. 2a; full serial optical slices of these structures can be viewed in Supplementary Movies 1 and 2). These topographically variable regions are morphologically consistent with similar structures observed in older literature[31,32]. However, the associated components or potential function of these protrusion-like structures has been largely unexplored[33]. Notably, the *Net1* mRNA localized within, and quite frequently at the tips of these structures (Fig. 2a; yellow arrowheads). Despite their smaller size, these keratinocyte structures appear morphologically similar to the in vitro cultured mesenchymal cell protrusions, where *Net1* and other protrusion-localized mRNAs have been previously studied[14–16]. These data thus raise the intriguing possibility that these poorly characterized keratinocyte structures reflect in vivo equivalents of in vitro protrusion-like structures and are analogously associated with localized mRNAs, including *Net1*.

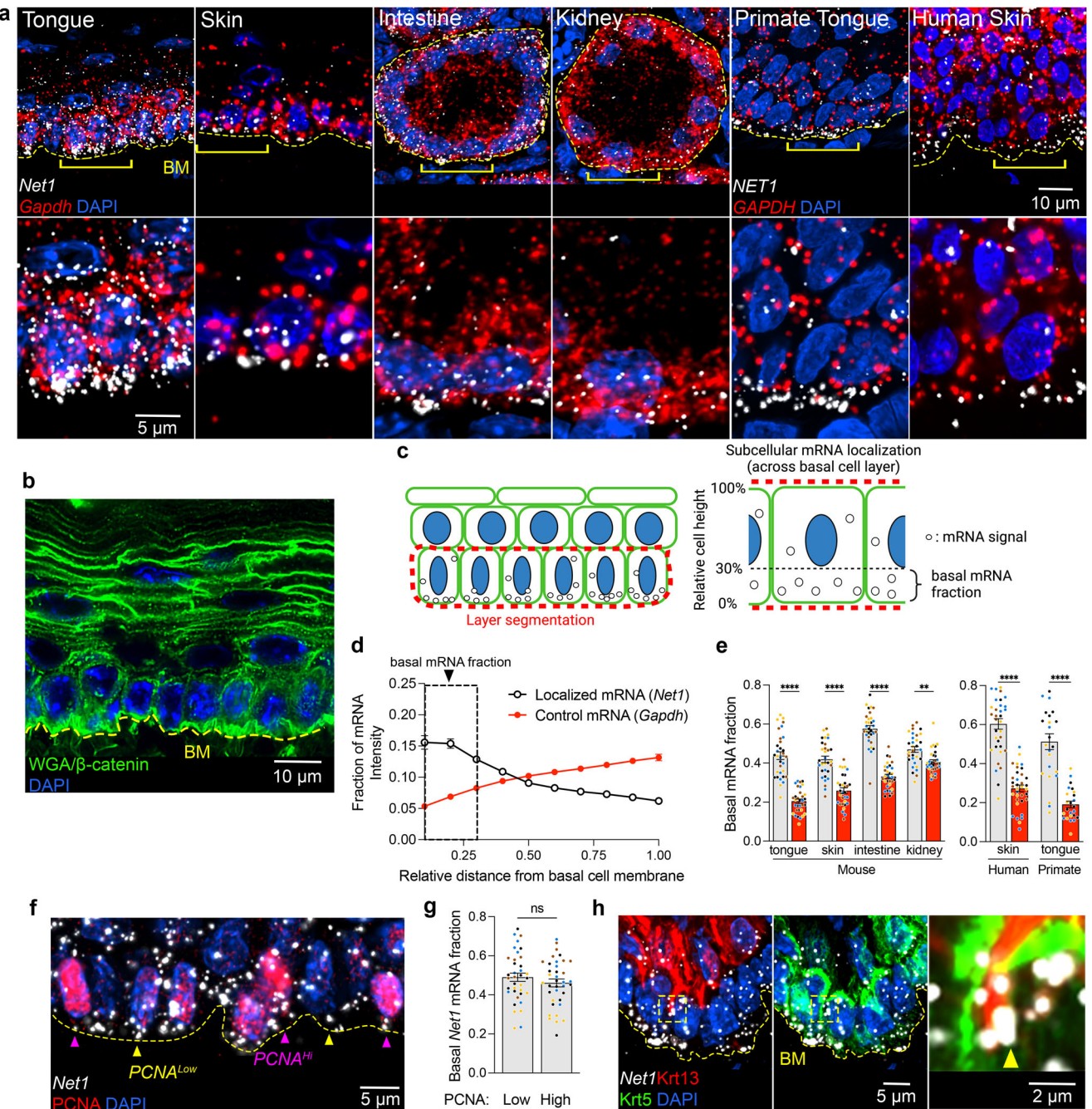

**Fig. 1 | Conserved basal localization of *Net1* mRNA across epithelial architectures and cellular contexts. a** Upper panels: Representative images of *Net1* and *Gapdh* mRNAs in mouse tongue, skin, intestine, and kidney as well as images of primate tongue and human skin. Dashed yellow line indicates the basement membrane (BM). Bottom panels: zoomed in regions around areas indicated by brackets. **b** Representative image of wheat germ agglutin (WGA) and β-catenin, as well as nuclei by DAPI in mouse tongue. Staining was used for segmenting the BM-attached keratinocyte layer which was then used to quantify mRNA distribution at a subcellular level. **c** Schematic depicting segmentation of BM-attached cell layer (red dashed line). Zoomed-in scheme to the right shows that the RNA amount found in the bottom 30% of the cell layer is defined as the 'basal RNA fraction'. (Created in BioRender. Mason, D. (2025) https://BioRender.com/ldh2jqb) **d** *Net1* and *Gapdh* mRNA fluorescent intensity in subsections starting at the BM to the most apical portion of the BM-attached cell layer. Boxed region indicates the subcellular region used to define the basal mRNA fraction. **e** *Net1 mRNA* (white bars) exhibits basal enrichment relative to *Gapdh* (red bars) across tissues and species. **f** Visualization of *Net1* mRNA and PCNA in mouse tongue. Yellow arrowheads indicate PCNA^Low and pink arrowheads indicate PCNA^Hi cells. **g** Basal fraction of *Net1* in PCNA^Hi vs PCNA^Lo cells. **h** Representative images of basal (Krt5+) and spinous (Krt13+) cells with an inset showing *Net1* mRNA localized to the basal tips of Krt13+ cells (yellow arrowhead). Line and bar graph data are measurements from individual tissue regions (ROIs) with mean ± SEM; $n = 32$ (**d**, **e**) or 36 (**g**) ROIs from $N = 4$ mice. Data from individual mice are represented with different colors. ** $P \leq 0.01$, **** $P \leq 0.0001$, ns non-significant by Brown–Forsythe and Welch ANOVA followed by Dunnett's T3 multiple comparison test (**e**), or unpaired two-tailed Student's *t*-test (**g**). Source data are provided as a Source Data file.

Protrusion-like basal membrane structures can also be visualized upon staining with WGA (Fig. 2b; serial optical slices can be viewed in Supplementary Movies 1 and 2). Aside from visualizing the interdigitations this staining also revealed a complex fiber network that spans the dermis and appears to connect with the protrusion-like structures at the DEJ (Fig. 2b; note that this image presents a projection of multiple optical slices to allow better visualization of the fiber network). To determine whether these structures are specific to oral

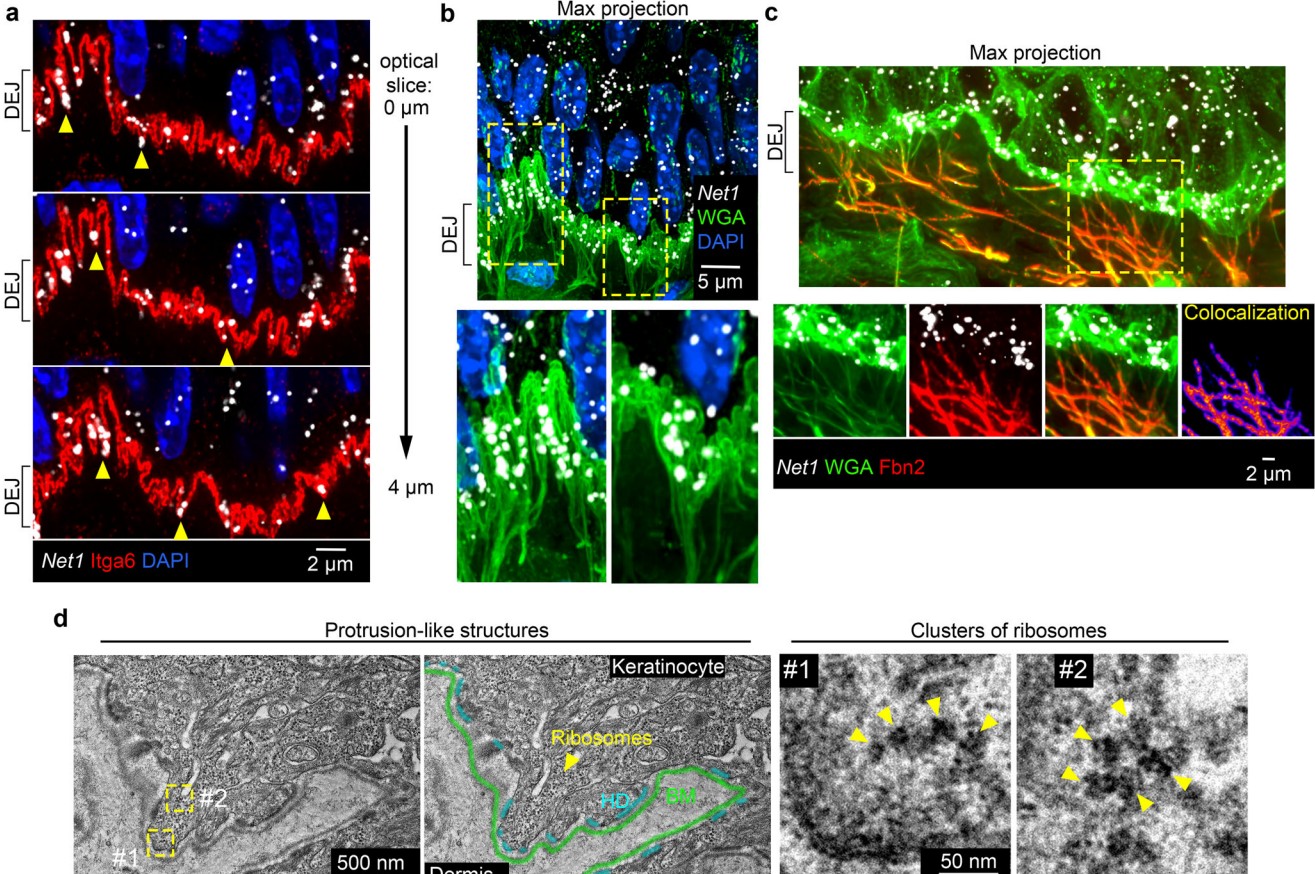

**Fig. 2 | *Net1* accumulates in translation-competent keratinocyte protrusion-like structures connected to microfibrils at the DEJ. a** Representative images of mouse tongue showing *Net1* mRNA at the DEJ. The basal keratinocyte cell membrane is visualized by Itga6 in sequential optical slices. Arrowheads indicate *Net1* mRNA at protrusion-like structures. **b** *Net1* and WGA staining of mouse tongue. Bottom panels: zoomed in regions, indicated by yellow boxes, highlighting fibers connecting with the basal keratinocyte cell membrane. Image is a max-intensity projection of ~3 um in the *z*-axis. **c** Mouse tongue section stained for *Net1* mRNA, WGA, and Fbn2. Bottom panels show individual channels and overlay from boxed region. Colocalization panel shows pixel intensity overlap between WGA and Fbn2 channels. **d** Transmission electron micrograph of a keratinocyte protrusion-like structure from mouse tongue. DEJ components, basement membrane (BM), and hemidesmosomes (HD), are indicated. Ribosomes are evident as electron dense particles. Yellow boxed areas are enlarged in the right panels to indicate strings of ribosomes, likely corresponding to actively translating polysomes.

epithelium we visualized the DEJ in sections of mouse tail skin (Supplementary Fig. 3). Staining for Itga6 revealed similar protrusion-like structures of the basal plasma membrane with local accumulation of the *Net1* mRNA (Supplementary Fig. 3a). Additionally, these areas were connected to WGA-positive dermal fibers (Supplementary Fig. 3b). Therefore, accumulation of *Net1* mRNA at basal keratinocyte protrusions that connect to a dermal fiber network is broadly observed in stratified epithelia.

To determine what these WGA-positive fibers correspond to, we surveyed various components of the DEJ and the dermal ECM. Co-detection with WGA indicated that these fibers are not enriched for hemidesmosome (Col17a1), basement membrane (laminin), or anchoring fibril (Col7) DEJ components (Supplementary Fig. 4a). Because these fibers form an extensive network below the DEJ, we hypothesized that they are part of the dermal ECM. However, detection of collagen I (Col1a1), the primary dermal ECM component, showed that these fibers are not enriched for Col1a1 except at regions proximal to the DEJ (Supplementary Fig. 4a). We discovered, though, that these fibers are strongly positive for fibrillin 2 (Fbn2), a major component of the microfibrils that are part of the elastic fiber network which allows transmission of mechanical forces across the dermis (Fig. 2c)[34–38]. Given the fact that stratified squamous epithelium is regularly mechanically deformed and that the microfibrils are essential

in resisting this deformation, it is tempting to speculate that keratinocyte protrusions are sites of high mechanical stress. Pertinent to this, targeting of protrusion-localized mRNAs in in vitro cultures is coordinated with cellular mechanical state and influenced by ECM properties in various settings[12,17]. Therefore, the DEJ could reflect a physiological setting where the mechanoresponsive localization of mRNAs is particularly relevant.

To assess whether the basal protrusion-like structures could be sites of mRNA translation we sought to determine whether ribosomes are present in these areas. For this, we visualized mouse tongue epithelium using transmission electron microscopy. We could observe within basal keratinocyte protrusions numerous ribosomes as electron dense particles with an average diameter of 24.31+/−3.97 nm (Fig. 2d, left panels). In some instances, clusters or strings of ribosomes were evident, likely corresponding to actively translating polysomes (Fig. 2d, right panels). As a complementary approach, we immunostained mouse tongue epithelium for ribosomal proteins of the small (Rps27) and large (Rpl23a) ribosomal subunits followed by Stimulated Emission Depletion (STED) super-resolution microscopy (Supplementary Fig. 4b). Proximity of Rps27 and Rpl23a signals within a radius of 100 nm or less would indicate a physical distance consistent with the presence of the two proteins in the context of a translating 80S ribosome. Indeed, STED imaging revealed several instances of putative 80S

complexes near the basal plasma membrane in keratinocyte protrusion-like structures (Supplementary Fig. 4b). The presence of ribosomes thus suggests that basal keratinocyte protrusions can support local mRNA translation.

## Localized Net1 mRNA maintains tissue and DEJ architecture

To broadly address the functional role of *Net1* mRNA localization in epithelial physiology, and protrusion-like structures specifically, we employed sequence blocking phosphorodiamidate morpholino oligos (PMOs). PMOs antisense to GA-rich regions of human protrusion-localized mRNAs have been shown to specifically prevent the localization of the targeted mRNA by blocking the formation of a transport-competent complex between the mRNA and the KIF1C kinesin[15,16,39]. To determine whether we can extend the use of this approach to mice, we designed PMOs that target the analogous GA-rich regions in the mouse *Net1* 3'UTR and tested them in in vitro cultured mouse NIH/3T3 fibroblasts by visualizing *Net1* mRNA using transcript-specific probes (Supplementary Fig. 5a–d). PMOs were either targeted to *Net1* GA-rich regions (PMOs #992 and #1016) or another downstream area (PMO #1620), or to an unrelated sequence (control PMO) (Supplementary Fig. 5e). Indeed, GA-targeting PMOs, but not other sequences, significantly prevented the localization of the *Net1* mRNA to the periphery of 3T3s without affecting another protrusion-localized mRNA, *Cyb5r3* (Supplementary Fig. 5f, g). Importantly, and consistent with prior reports in human cells, the effect on *Net1* mRNA localization was not accompanied by any detectable change in the amount of *Net1* mRNA (Supplementary Fig. 5h)[16]. We also measured the enrichment of various mRNAs within isolated 3T3 protrusions and observed that Net1 PMO delivery affected solely the enrichment of the *Net1* mRNA while all other detected transcripts were unaffected (Supplementary Fig. 5i). We further validated that Net1 PMOs #992 and #1016 can similarly disrupt *Net1* mRNA localization in an immortalized keratinocyte cell line in vitro (Supplementary Fig. 6a, b), again without affecting *Net1* mRNA levels or the amount of Net1 protein produced (Supplementary Fig. 6c–e). These data confirm the broad conservation of the mRNA trafficking mechanisms to protrusions, and the applicability of PMO delivery as a specific tool to modify mRNA distributions in diverse cell types.

To assess the physiological importance of *Net1* mRNA localization in vivo we intradermally injected the two *Net1* localization altering PMOs or two control PMOs (one targeting GFP and one corresponding to a scrambled Net1 #992 sequence) into mouse tongues. PMOs were administered every 2 days over 6 days (Fig. 3a). We confirmed, by small RNA in situ hybridization, that PMOs reached and were taken up by the cells in the epithelium (Fig. 3b). Significantly, when we measured *Net1* mRNA distribution in injected tissues, there was a substantial reduction in the accumulation of *Net1* mRNA in the basal fraction of BM-attached epithelial cells upon administration of the GA-rich region targeting PMOs (Fig. 3c, d). Therefore, PMOs can prevent the subcellular accumulation of the endogenous *Net1* mRNA in epithelial tissues in vivo.

Further phenotypic analysis revealed prominent defects in epithelial architecture detected by H&E staining (Fig. 3e). Specifically, the epithelium of Net1 PMO-treated tongue epithelia was thinner in comparison to control treated tissues (Fig. 3f). Such changes in epithelial architecture can be brought about by defects in cellular differentiation[40,41]. Visualization of canonical markers of the basal and spinous compartment, Krt5 and Krt13, respectively, revealed that Net1 PMO-treated tongue contained significantly more Krt5/Krt13+ double positive cells in the basal layer (Fig. 3g–i). This suggests either spontaneous differentiation of Krt5+ cells or alternatively a defect in cell detachment from the basement membrane.

Given the fact that *Net1* mRNA localizes to protrusion-like structures at the DEJ, we investigated whether altering *Net1* localization from the basal surface affects DEJ organization. Indeed, Net1 PMO-

treated epithelium exhibited drastic flattening of protrusion-like structures formed by basal keratinocytes (Fig. 3j; serial optical slices can be viewed in Supplementary Movie 3). We quantified this change by measuring topographical variation of the basal plasma membrane using optical slices of Itga6-stained tissues and deriving a 'DEJ variation' metric, a value that reflects the degree to which basal keratinocytes exhibit protrusion-like structures (see Supplementary Fig. 7 and "Methods" for details). Using this metric, DEJ variation was halved in Net1 PMO-treated epithelia (Fig. 3k). As shown above, basal protrusions appear to be connected to Fbn2+ fibers. We thus asked whether altering *Net1* localization has broader effects on the microfibrils near the DEJ. Consistent with this, the volume of Fbn2+ fibers was significantly reduced in regions proximal to the DEJ (Fig. 3l, m). This result could reflect ECM remodeling near the epithelium but may also be due to decreased antibody-epitope accessibility on individual fibers due to altered mechanical strain, which can lead to changes in fibrillin domain folding[42]. Regardless, it is notable that dysregulation of the DEJ during aging or in genetic diseases, including ones caused by mutations in fibrillins, has strong negative implications on epidermal physiology[33,34,38,43]. Altogether, these results demonstrate that altering the basal localization of a single mRNA, *Net1*, is sufficient to drastically influence keratinocyte physiology and tissue homeostasis, potentially by altering interactions of the basal epithelium with the DEJ.

## mRNA targeting controls Net1 protein activity in vivo

To understand the mechanism through which altering *Net1* mRNA localization affects keratinocyte physiology, we considered that *Net1* mRNA location might affect the activity of the encoded Net1 protein as a RhoA GEF. Net1 is quite unique among RhoA regulators in that it is controlled by nucleo-cytoplasmic trafficking[44,45]. While RhoA activation largely occurs at the plasma membrane, Net1 can be imported and sequestered in the nucleus, providing an "off-switch" for Net1-dependent RhoA activation[16,19]. *Net1* mRNA location controls Net1 nuclear import versus cytoplasmic retention by a partner-selection mechanism in mesenchymal cells in vitro. Specifically, translation of Net1 from protrusion-localized mRNA favors its interaction with a plasma membrane scaffold and promotes RhoA activation by Net1[16]. To determine whether this mRNA location-dependent regulation operates in the tongue epithelium in vivo, we looked at the subcellular distribution of the Net1 protein. We first validated the specificity of the Net1 antibody, by immunofluorescence staining of immortalized keratinocytes upon Net1 knock down with two different siRNAs (Supplementary Fig. 8a, b). We then observed Net1 protein distribution in tongue sections. Net1 protein signal was primarily detected in the layer of BM-attached keratinocytes (Fig. 4a), consistent with the basal cell specific expression pattern of *Net1* mRNA, mentioned above (Supplementary Fig. 2e). This further supports the specificity of Net1 protein detection. We note that while Net1 protein is primarily nuclear and less cytoplasmic in cultured keratinocytes, in in vivo tissue Net1 is primarily cytoplasmic and specifically accumulating in the cell cortex (compare Fig. 4a and Supplementary Fig. 8a). While the basis for this differential regulation is still unclear, this has been also observed in other studies[19,46,47]. Additionally, Net1 is observed throughout the keratinocyte cortex and does not accumulate solely at the basal plasma membrane (Fig. 4a). This observation is consistent with in vitro results showing that Net1 protein diffuses away from the site of translation and can be either retained broadly in the cytoplasm or imported into the nucleus depending on the location of its synthesis[16]. To quantify changes in Net1 protein distribution we measured its cortical/non-cortical intensity (Fig. 4a–c). Net1 PMO-treated epithelium showed a reduced cortical Net1 accumulation (Fig. 4d), indicating that *Net1* mRNA location also controls Net1 protein distribution in vivo.

A reduction in cortical Net1 amount would be predicted to result in reduced activation of the RhoA GTPase. Given that it is technically difficult to directly measure RhoA activity in tissues, we examined the

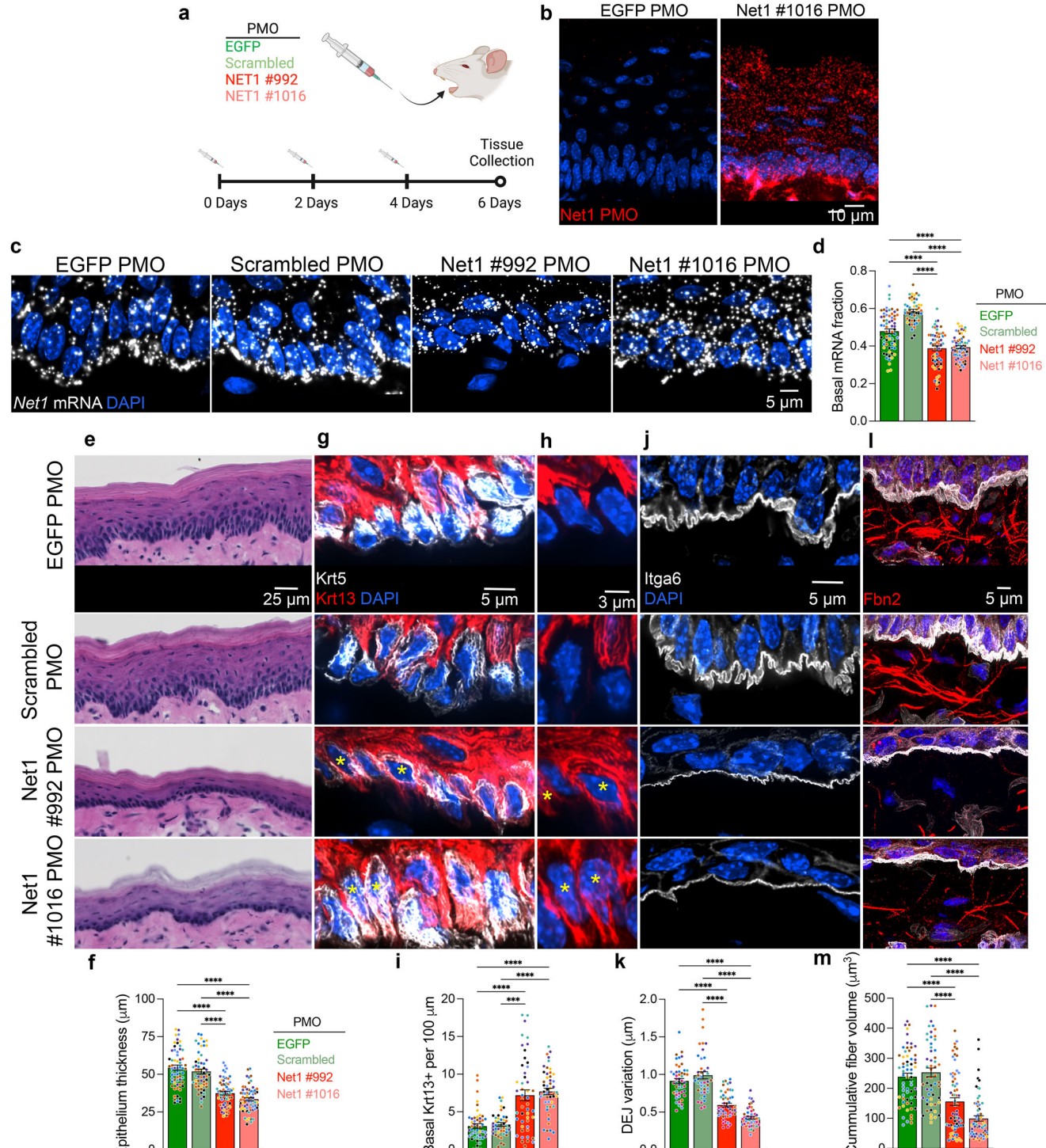

**Fig. 3 | *Net1* mRNA localization controls epithelial homeostasis and keratinocyte interaction with the DEJ. a** Schematic of PMO injection protocol into the ventral side of mouse tongues using sequences targeting *EGFP* or the *Net1* 3'UTR at positions #992 and #1016, as well as a scrambled version of #992. (Created in BioRender. Mason, D. (2025) https://BioRender.com/b2e2dfg) **b** PMO uptake into the tongue epithelium was visualized using Net1 #1016 PMO-specific probes. **c** Visualization of *Net1* mRNA distribution in PMO-injected tongues. **d** Basal *Net1* mRNA fraction in the BM-attached keratinocyte cell layer of PMO-treated tissues. **e** Visualization of overall oral epithelium architecture using H&E on ventral portion of the tongue. **f** Measurement of tongue epithelial thickness. **g** Visualization of basal (Krt5) and spinous (Krt13) keratinocytes in the basal compartment of tongue epithelium. **h** Select areas from (**g**). Krt13 channel is only shown to highlight examples

of Krt5+/Krt13− basal cells in control PMO-treated tongues and Krt5+/Krt13+ in Net1 PMO-treated tongues. Asterisks indicate double positive cells. **i** Number of Krt5+/Krt13+ cells in the basal compartment normalized to DEJ length. **j** Visualization of DEJ using Itga6 in PMO-treated mouse tongues. **k** Measurement of DEJ topographical variation. **l** Visualization of Fbn2+ fibers in PMO-treated tongues. **m** Quantification of DEJ proximal Fbn2+ fiber volume. Bar graph data are measurements from individual tissue regions (ROIs) with mean ± SEM; $n = 42$–$64$ ROIs from $N = 7$–$8$ mice per condition. Data from individual mice are represented with different colors. *** $P \leq 0.001$ and **** $P \leq 0.0001$ by Brown−Forsythe and Welch ANOVA followed by Dunnett's T3 multiple comparison test (**d**) or Kruskal−Wallis ANOVA followed by Dunn's multiple comparison tests (**f**, **l**, **k**, **m**). Source data are provided as a Source Data file.

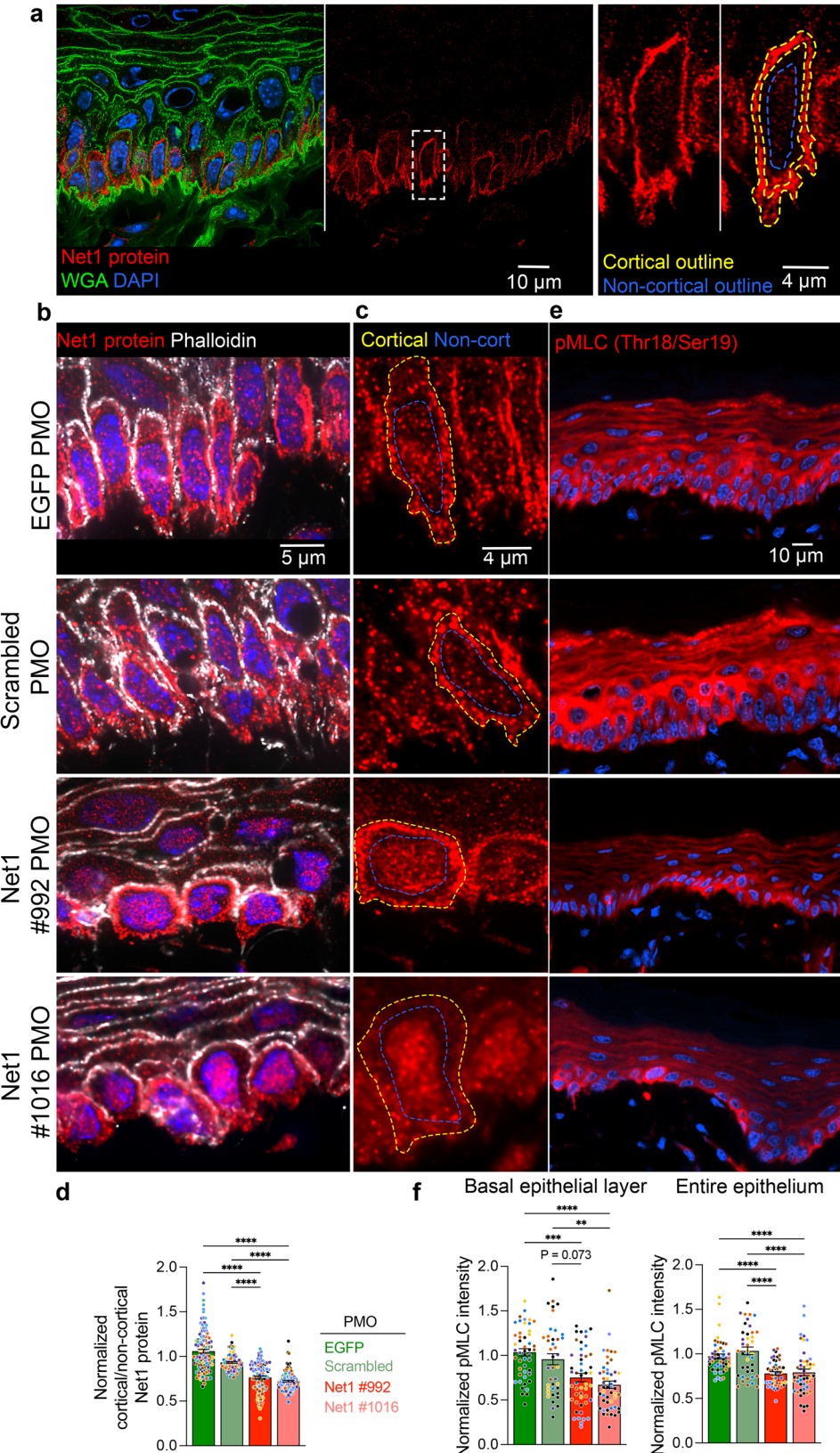

phosphorylation of myosin light chain (pMLC), a major downstream RhoA target[48]. pMLC levels were visualized using a phospho-specific antibody under conditions that allow specific detection (Supplementary Fig. 9). Quantification of pMLC levels in Net1 PMO-injected tongue epithelium revealed that altering *Net1* mRNA localization reduced pMLC in both basal keratinocytes and more broadly across the entire epithelium (Fig. 4e, f). Therefore, preventing the basal localization of the *Net1* mRNA leads to altered Net1 protein distribution and a reduction in RhoA-pMLC signaling. Given that this pathway is well known to control cytoskeletal tension and cell attachment, it is likely that it reflects the mechanism linking *Net1* mRNA targeting to DEJ structure and epithelial organization.

**Fig. 4 | *Net1* mRNA localization regulates Net1 protein distribution and activity in vivo. a** Visualization of Net1 protein and WGA in mouse tongue keratinocytes. Left panel: overlay. Middle panel: Net1 channel showing predominant expression in basal keratinocytes. Right panels: zoom in of boxed region. Dashed yellow and blue lines indicate cortical and non-cortical regions, respectively, used for Net1 distribution measurements. **b** Visualization of Net1 protein with phalloidin as a marker of the cell cortex in PMO-treated tissues. **c** High magnification visualization of Net1 protein where the cell cortex and non-cortical region are indicated by yellow and blue broken lines, respectively. **d** Quantification of Net1 protein distribution (ratio of cortical to non-cortical mean intensity). **e** Visualization of MLC2 phosphorylation (pMLC) in PMO-treated mouse epithelium. **f** Quantification of mean pMLC2 fluorescent intensity for basal keratinocytes (left) and the entire oral epithelium (right). Bar graph data are individual measurements with mean ± SEM; $n = 105–120$ cells (**d**) or 40–48 ROIs (**f**) from $N = 7–8$ mice per condition. Data from individual mice are represented with different colors. ** $P \leq 0.01$, *** $P \leq 0.001$, **** $P \leq 0.0001$ by Kruskal–Wallis ANOVA followed by Dunn's multiple comparison test. Source data are provided as a Source Data file.

## Basal keratinocyte cytoskeletal state directs DEJ morphology

To test this idea, we tried to independently address whether changes in RhoA signaling within basal keratinocytes control their interaction with the DEJ. For this, we used a genetic system for cell-type specific RhoA activation[49,50]. Krt14-rtTA mice were crossed with a genetic knock-in of a constitutively active RhoA GEF, ArhGEF11 (ArhGEF11$^{CA}$), under the control of a tetracycline-inducible promoter (Fig. 5a), to achieve RhoA activation specifically in basal keratinocytes. Doxycycline injection led to basal keratinocyte-specific overexpression of ArhGEF11$^{CA}$ in mouse tongue, visualized by an HA-tag, as early as 10hrs post-injection (Fig. 5b). ArhGEF11$^{CA}$-expressing cells exhibited increased cortical pMLC and filamentous actin (phalloidin staining) indicative of RhoA-induced cytoskeletal tension (Fig. 5b).

We examined tongue tissue architecture in these mice, comparing them to mice with the ArhGEF11$^{CA}$ cassette that were not crossed with Krt14-rtTA mice (WT/ArhGEF11$^{CA}$). We focused our quantitative analysis on DEJ regions where all of the basal keratinocytes express the HA-tagged ArhGEF11$^{CA}$. Remarkably, this short-term induction of RhoA signaling was sufficient to significantly increase the prominence of keratinocyte protrusion-like structures indicated by an increase in DEJ topographical variation (Fig. 5c, d). Concomitantly, DEJ proximal Fbn2+ fibers became more prominent and displayed a higher overall volume in regions proximal to the epithelium (Fig. 5e, f). Given the short induction time we favor the idea that increased fibrillin detection reflects, at least partly, increased epitope accessibility through tension-induced unfolding, rather than solely an increase in their absolute amount.

A unique feature observed upon ArhGEF11$^{CA}$ expression was the occasional appearance of Itga6 strands co-localizing with Fbn2+ fibers far below the overall level of the basal plasma membrane (Fig. 5g). The fact that these do not appear under normal conditions suggests that they are formed upon the supraphysiologic tension induced under our experimental setting. We envision that fibrillin resistance to keratinocytes' tensile forces eventually leads to fiber retraction away from the epithelium. In this scenario the residual tension in fibrillin fibers pulls portions of the plasma membrane below the ordinary DEJ (Fig. 5g). The tight colocalization of Itga6 with fibrillin suggests the existence of stable mechanical coupling between the keratinocyte plasma membrane and microfibrils. These data provide orthogonal evidence that changes in RhoA activity, and cytoskeletal tension, is sufficient to alter keratinocyte connections with the DEJ. Therefore, the RhoA pathway is the likely mediator of the ultrastructural changes at the DEJ in response to altering *Net1* mRNA localization. Overall, the evidence presented here reveals mRNA localization as an unappreciated level of post-transcriptional regulation that controls the mechanical coupling of the epithelium with the underlying connective tissue, which broadly affects tissue physiology.

## Discussion

Post-transcriptional mRNA regulation has well known roles both in normal physiology as well as in disease, exemplified for instance by miRNA-mediated regulation during development[51], or the therapeutic modulation of splicing patterns[52]. Targeting of mRNAs to specific subcellular domains has emerged as another widespread level of post-transcriptional mRNA control with the potential to spatially coordinate protein-protein interactions at a subcellular level and control protein function[4,7,16]. Nevertheless, the contribution of mRNA localization to mammalian physiology has remained largely unexplored. Here, we demonstrate that the subcellular localization of an individual mRNA, *Net1*, is an important regulator of epithelial tissue homeostasis. This observation underscores the importance of this post-transcriptional regulatory mechanism in epithelial tissue physiology where ~15–30% of the transcriptome is differentially polarized[2,3] and sensitive to physiological stimuli[2].

Notably, we show that the basally localized *Net1* mRNA is associated with protrusion-like structures of keratinocytes that contact a mechanosensitive network of dermal microfibrils. We speculate that these underappreciated structures at the DEJ may be mechanical connections necessary for long range force transmission between the epidermis and dermis. Targeting of the *Net1* mRNA at these DEJ structures is necessary for their formation and its loss results in remodeling of the dermal ECM. Our evidence further suggests that these effects are mediated through basal cell autonomous activation of RhoA. Indeed, Net1-mediated activation of RhoA is necessary for collagen I remodeling in breast tissue[53] as well as basement membrane breakdown during embryonic development[46]. The findings presented here indicate that such RhoA-regulated cell-ECM interactions are mediated at least partly by localized translation of the *Net1* mRNA. Interestingly, *Net1* mRNA localization is itself mechanosensitive, being affected by the properties of the ECM[12,17], implying the existence of feedback regulation that maintains an RNA-dependent mechanical homeostasis of the DEJ. This maintenance of homeostasis extends to keratinocyte function, including differentiation, which is regulated by cell-autonomous and tissue-wide variation in mechanical state[40,41].

Alterations in DEJ components, including breakdown of the microfibrillar network and a morphological attenuation of keratinocyte protrusions, occurs during aging[33,43]. Additionally, mutations that affect the BM or fibrillins have deleterious effects on epidermal homeostasis and have been implicated in a number of genetic diseases and malignancies[30,38,54]. These include mutations that cause delamination of the epidermis and increased propensity for tumor development[55,56]. We speculate that changes in basal localization of *Net1*, and likely other protrusion mRNAs, might contribute to this disease- and aging-associated mechanical uncoupling of the epidermis. Given that Net1 is widely expressed and consistently localized in a variety of tissues it is possible that its unique regulatory mechanism is utilized for maintenance of epithelium-ECM connections in additional tissues. Further understanding of the dynamic control and functional contributions of such events occurring at tissue interfaces will provide exciting new directions and context for studying RNA transport and local regulation in mammalian physiology.

## Methods

### Animal handling and tissue collection

All experiments involving animals were performed in accordance with guidelines set by the Institutional Animal Care and Use Committees of the National Cancer Institute (National Institutes of Health, Bethesda, MD) and Duke University (Durham, NC; Protocol #: A255-23-12). All mice (4–12 weeks) were housed in sterile filter-capped cages, fed and watered ad libitum, in a 12:12 light-dark cycle animal facility maintained at 72 °F ± 3 °F with an ambient humidity of 50% ± 15%.

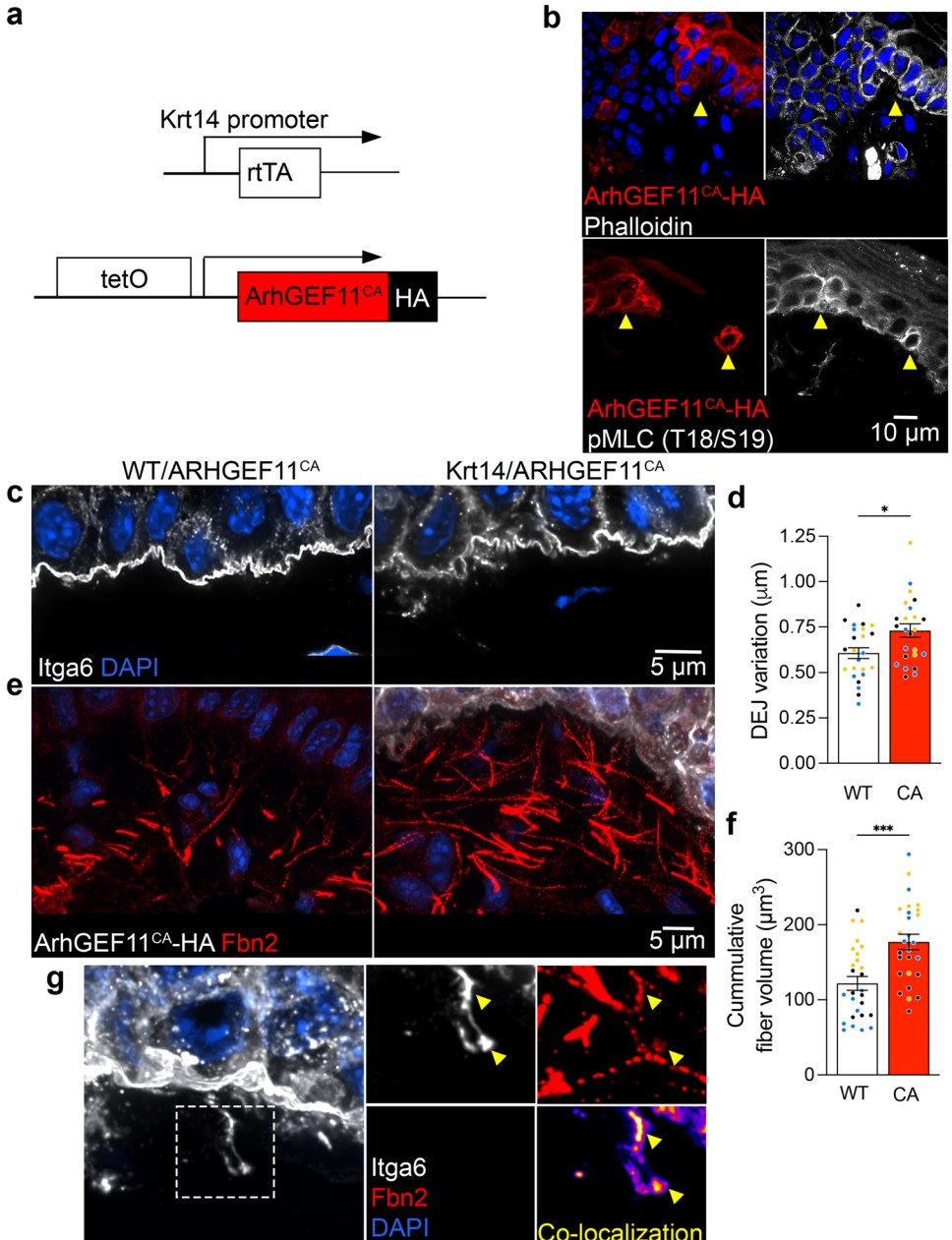

**Fig. 5 | RhoA activation in basal keratinocytes remodels the DEJ. a** Schematic of ArhGEF11$^{CA}$-HA expression under the control of tetracycline-inducible promoter where the reverse tetracycline-controlled transactivator is expressed in basal epithelial cells (Krt14-rtTA). **b** Visualization of the actin cytoskeleton (phalloidin) and pMLC in ArhGEF11$^{CA}$-HA-expressing cells. Mosaic expression of ArhGEF11$^{CA}$ is indicated by yellow arrowheads. **c** Visualization of the DEJ by Itga6 in WT or ArhGEF11$^{CA}$ mice. **d** Quantification of DEJ variation in knock-in mice. Measurements in ArhGEF11$^{CA}$ (CA) mice were made in regions with continuous HA expression in basal keratinocytes across an analysis ROI. **e** Visualization of HA and Fbn2 at the DEJ of WT or ArhGEF11$^{CA}$ mice. **f** Quantification of Fbn2 volume proximal to the DEJ. **g** Left panel: example of an ArhGEF11$^{CA}$-expressing cell exhibiting extension of the plasma membrane below the DEJ after supraphysiologic induction of RhoA activation. Plasma membrane is stained through Itga6. Right panels: Boxed region showing also Fbn2 staining and co-localization between the two channels. Yellow arrowheads indicate areas of high Itga6-Fbn2 co-localization. Bar graph data are measurements from individual ROIs with mean ± SEM; $n = 24$ (**d**) or 27 (**f**) ROIs from $N = 3$ mice. Data from individual mice are represented with different colors. * $P < 0.05$, *** $P < 0.001$ by unpaired two-tailed Student's $t$-test. Source data are provided as a Source Data file.

For injections, custom octaguanidine dendrimer conjugated PMOs (Vivo-Morpholino, GeneTools) that target *EGFP* or *Net1* mRNA were synthesized (Supplementary Table 1). PMOs (5 nmol/inj) and fluorescently labelled dextran (2.5 µg/inj.; Molecular probes; D1976) were mixed in isotonic sterile saline. 10 µL of the solution was subcutaneously injected in the ventral portion of the tongues of anesthetized male and female C57BL/6/NCr mice (Charles River; age 10–12 weeks). Anesthesia was performed by brief exposure to 3% isoflurane followed by intraperitoneal injection of 100 mg/kg ketamine and 10 mg/kg xylazine. The tongues were injected with PMOs every

two days for a total of three injections. Mice were sacrificed for tissue collection 6 days after the initial injection.

Mice overexpressing ArhGEF11$^{CA}$-HA in Krt14+ keratinocytes were generated using previously described transgenic mouse models[50,57,58]. Briefly, mice with the TRE-Arhgef11-HA transgene[50] were crossed with FVB-Tg(KRT14-rtTA) (Jackson Laboratory Strain #: 008099)[57,58]. 4-week-old TRE-Arhgef11/Krt14-rtTA mice (4 males and 2 females) were IP-injected with 25 mg/kg doxycycline (Sigma) 10 h before sacrifice and tissue collection. Transgene expression was validated using an HA-tag specific antibody (Roche; 11867423001).

Tissue was collected from male and female euthanized mice (age 4–12 weeks) washed gently in PBS then flash frozen in optimal cutting temperature compound (OCT; Fisher Healthcare; 4585). Tissue embedded OCT blocks were cut 8–10 um thick on a Leica cryostat microtome CM1860. Tissues were collected on positively charged slides (Rankin; 20290 W) then stored at −80 °C. The gross tissue morphology of each tissue was visualized by H&E staining (abcam; ab245880).

## Mouse keratinocyte cell line generation

Mouse keratinocytes were isolated from tail skin of male mixed background mice (FVB/N and C57) by physical separation of the epidermis and dermis from the hypodermis. Tail skin was sterilized with 10% v/v betadine/iodine in PBS. Excised tissue was digested overnight at 4 °C using 2 U/mL of dispase II (StemCell; 07913) diluted in 0.25% trypsin without EDTA (Gibco; 15050065). After digestion the tissue was minced and mixed with media and antibiotics. Keratinocytes were then isolated from the digest by sequential straining using 100 μm and 40 μm cell strainers. Cells were cultured for one passage using EpiLife medium with 60 μM calcium (Gibco, Waltham, MA, MEPI500CA), supplemented with Human Keratinocyte Growth Supplement (HKGS, Life Technologies, S0015), mouse EGF (10 ng/ml, R&D Systems, Minneapolis, MN, 2028EG200), and Y-27632 compound (10 μM, Tocris Bioscience, Bristol, United Kingdom, 12-541-0). Keratinocytes were then immortalized by infecting with lentiviruses expressing SV40 Large T Antigen (Addgene Plasmid #170255). Cells were selected with Hygromycin (200 μg/mL) for two days and then cells were passed for two passages in EpiLife and described supplements. Then cells were cultured in DMEM (Sigma-Aldrich Inc) containing 10% fetal bovine serum (FBS) (Sigma-Aldrich Inc), antibiotic/antimycotic solution (Sigma-Aldrich Inc), mouse EGF (10 ng/ml, R&D Systems, Minneapolis, MN, 2028EG200), and Y-27632 compound (10 μM, Tocris Bioscience, Bristol, United Kingdom, 12-541-0), for at least two passages before experiments. Lentiviruses were produced by transfecting Lenti-X 293 T cells with pMD2.VSVG (Addgene plasmid #12259) and psPAX2 (Addgene plasmid #12260). Lenti-X 293 T cells were obtained from Takara Bio and cultured in DMEM (Sigma-Aldrich) containing 10% fetal bovine serum (Sigma-Aldrich).

## Immunofluorescence and tissue staining

Fresh frozen sections or cultured cells were fixed in 4% formaldehyde (Sigma; F8775) for 15–30 min at RT washed 3 × 2 min in PBS and permeabilized in 0.2% Triton-X-100 for 5–20 min. Sections were washed in PBST and blocked for 1 h in 3% bovine serum albumin (BSA) and 2% goat serum (GS) in PBS. Antibodies were diluted in blocking buffer and incubated overnight (Supplementary Table 2). Samples were washed 3 × 2min in PBST and incubated with fluorophore conjugated secondary antibodies corresponding to the host species (Supplementary Table 2; 1:500 in blocking buffer). Samples were washed with PBST 3 × 2 min then counterstained with a fluorophore conjugated variation of wheat germ agglutin (Invitrogen; W11261; 10 ug/mL), phalloidin (Invitrogen; A12379), and/or DAPI. Tissue samples were mounted using #1.5 coverglass (Fisher; 12544D) with either prolong glass or diamond mounting medium (Invitrogen; P36980 or P36962) and allowed to set overnight at room temperature.

Phosphorylated-myosin light chain 2 (Thr18/Ser19) was detected in tissue samples using a modified immunofluorescence protocol. Briefly samples were fixed in 4% formaldehyde for 3 h at RT, washed 3 × 5 min in TBS then treated with 1% SDS diluted in TBS for 20 min at RT with regular agitation. Samples were washed thoroughly in 3 × 5 min in TBST (0.1% Tween-20). Samples were blocked in 3%BSA and 2% GS in TBS for 1 h. pMLC specific rabbit mAb (Cell signaling; 95777) was diluted in blocking buffer and incubated at 4 °C overnight. Staining was completed as described above with TBS instead of PBS.

Specificity of the pMLC antibody was validated by pre-treating tissues with 4000 U λ phosphatase (NEB; P0753S) diluted in 1 × NEB-uffer for protein phosphatases and 1 mM $MnCl_2$ overnight at 30 °C.

Samples were washed thoroughly in TBST before proceeding with the staining protocol described above.

## Cell culture

NIH/3T3 mouse fibroblast cells (ATCC, cat# CRL-1658) were grown in Dulbeco's Modified Eagle Medium (DMEM; Gibco; 11995-065) supplemented with 10% calf serum (Cytiva; SH30087.04), sodium pyruvate, and penicillin/streptomycin (Gibco; 15140122). Immortalized mouse keratinocytes were cultured in DMEM supplemented with 10% Fetal Bovine Serum,10 ng/mL mouse EGF, and 10 μM Y27632. Cells were cultured at 37 °C, 5% $CO_2$ and passaged with either 0.05% (3T3) or 0.25% (keratinocytes) trypsin with EDTA (Gibco; 25300054 or 25200056). Cells used in this study have tested negative for mycoplasma.

For PMO delivery, cells were grown to 70% confluence in 12 well plates then transfected with the indicated PMOs at a concentration of 15 μM in basal media. For small interfering RNA (siRNA) experiments cells were cultured to 50–70% confluency in 6 or 12 well plates then transitioned to antibiotic free media. Cells where transfected with the indicated siRNA (Qiagen; final concentration 40 nM; Supplementary Table 1) which were pre-complexed with Lipofectamine RNAimax (Invitrogen; 13778075) transfection reagent according to the manufacturer's instructions. Cells were assayed for protein expression 48 h after transfection.

## Imaging

Fixed samples were imaged on a Nikon Eclipse Ti2-E inverted microscope with a Yokogawa CSU-X1 spinning disk confocal scanner unit and operated using NIS-Elements software (5.21.03, Build 1489). Images were acquired using 20× (Plan Apo 20×; NA = 0.75; WD = 1000 μm), 40× (Plan Fluor 40; Oil; NA = 1.30; WD = 240 μm), 60× (Apo 60× λS DIC N2; Oil; NA = 1.40; WD = 140 μm), or 100× (Apo TIRF DIC N2; Oil; NA = 1.49; WD = 120 μm) objectives and a Hamamatsu ORCA-Fusion BT Gen III back-illuminated sCMOS cameras. Images were denoised and deconvolved using the Richardson-Lucy algorithm in the NIS-Elements analysis software (5.21.03, Build 1489).

Whole sample brightfield images of H&E-stained samples were taken using a Zeiss Axio Scan.Z1 (Zeiss) equipped with a Hamamatsu OrcaFlash 4.0 camera using 10× and 20× objectives. Images were acquired using the Carl Zeiss Zen 2.3 software and analyzed using the HALO image analysis platform (Indica Labs).

For Stimulated Emission Depletion (STED) microscopy, images were recorded by STEDYCON (Abberior Instruments) assembled on Eclipse Ti2 inverted microscope (Nikon Inc.) and 100×, NA 1.45 Plan Apo objective. Avalanche photodetectors (650–700 nm; 575–625 nm; 505–545 nm; DAPI detection) were used to detect the signals. Browser-based control software (Abberior Instruments) was used to generate STED images. Excitation lasers for STAR RED and STAR ORANGE were run with ~10% laser power and depleted with the STED laser at 97.88% and 100%, respectively. Signals were detected within a 7 ns gate, using a pinhole size 32–64 μm and a pixel size of 10 nm.

## Transmission electron microscopy

Mouse tongue tissue was fixed in cacodylate buffer (0.1 M, pH7.4, CB) containing 2.5% glutaraldehyde (Sigma-Aldrich; G5882) and 0.5% formaldehyde (Electron Microscopy Sciences; 15686) for 2 h at room temperature and rinsed two times for 10 min in CB. Samples were postfixed in CB containing 1% osmium tetroxide (Electron Microscopy Sciences; 19100) in for 1 h, rinsed two times for 10 min in CB, and incubated in CB containing 0.5% of tannic acid (Electron Microscopy Sciences; 21700) for 30 min. After rinsing once in cold acetate buffer (0.1 N) for 10 min, the samples were pre-stained with in acetate buffer with 1% uranyl acetate (Electron Microscopy Sciences; 22400) for 1 h, rinsed twice in acetate buffer for 10 min each, dehydrated in graded ethanol series (30%, 50%, 70%, 95%, 100%) and propylene oxide, and embedded in EMbed-812 resin (Electron Microscopy Sciences; 13940).

Ultrathin (70 – 80 nm) serial sections were sectioned, placed on the formvar-coated copper grids (Electron Microscopy Sciences; FF2010-Cu-25), and stained with 1% uranyl acetate (in 50% ethanol) and Reynold's lead citrate. Imaging was performed using a FEI Tecnai T12, operating at 80 kV.

## Basal mRNA quantification

To calculate the 'basal mRNA fraction', maximum projections of ~1 μm thick optical sections of stained tissue were used. The BM-attached cell layer was manually segmented using WGA and β-catenin staining to guide segmentation. A custom script (MATLAB Mathworks; R2021b-R2023b) was used to separate the segmented layer into 10 evenly divided bins from the basal to apical side of a layer, and to measure mRNA fluorescent intensity in each bin after a uniform background subtraction. The fraction of mRNA intensity in each bin was then calculated per layer and used to measure relative fraction of basal mRNA (bottom 30% of a segmented layer). Analysis script is provided as a supplementary file (Supplementary Code 1).

## Net1 protein distribution

Net1 protein distribution was calculated on a per cell basis by manually segmenting basal keratinocytes in the mouse tongue using the FIJI(2.16.0/1.54 g) image analysis platform[59]. Segmentation was performed on individual optical slices taken with a 100× objective described above. Cortical Net1 protein mean intensity was measured in the peripheral 15% of the cell then normalized to the most central 65% of the segmented cell. Analysis script is provided as a supplementary file (Supplementary Code 2).

## DEJ variation analysis

To measure the complex protrusion-like structures at the DEJ we used a metric that reflects DEJ topographical variation (the process is schematically shown in Supplementary Fig. 7). Briefly, using FIJI(2.16.0/1.54 g), a segmented line ROI (120 pixels wide (7.8 μm)) was drawn so that it is centered on the basal cell membrane visualized through Itga6 staining. Itga6 fluorescent intensity was automatically thresholded and skeletonized. The coordinates of the basal cell membrane were estimated for each x-position using the skeletonized image. The x- and y- coordinates from these measurements were then used to create a 2D plot profile from which the absolute distance between the basal cell membrane and the center of the segmentation line was measured for each x-coordinate. Distances were summed and normalized to the number of x-positions measured to yield a single DEJ variation metric. This derived 'DEJ variation' metric is analogous to 'Average roughness (Ra)', a common measurement of surface texture[60]. Analysis script is provided as a supplementary file (Supplementary Code 3).

## Fibrillin 2+ quantification analysis

Fibrillin 2+ fiber volume was measured using 4 μm thick optical stacks acquired from epidermis proximal dermal areas (~50 μm$^2$) using a 100× objective. Imaris (Bitplane; v9.9.0) was used to generate Fbn2+ fiber volumes by thresholding Fbn2 fluorescence intensity equally between samples within a given experiment. Morphological filters were then equally applied across conditions to remove small spherical volumes that were not consistent with fiber segments. The cumulative fiber volume was then measured from all extracellular Fbn2+ fiber volumes in each image.

## Peripheral distribution index analysis

FISH images of in vitro cultured cells were analyzed using a previously published MATLAB script[61]. Briefly, the peripheral distribution index (PDI) of individual RNAs was measured by calculating the difference between the geometric centroid of a cell and the intensity weighted centroid of an RNA. This value is then normalized to a hypothetical uniformly distributed RNA. This ratio represents the relative peripheral distribution of an RNA in the context of a cell's geometry. RNAs are considered peripheral if PDI values are greater than 1, homogenous if equal to 1, and perinuclear if less than 1.

## RNA detection

Detection of RNA in tissues sections was performed using the RNAscope Multiplex Fluorescent Reagent kit v2 (ACD; 323100). Tissues were first pre-processed as follows: fresh frozen tissues were fixed for 45 min in 4% formaldehyde (Sigma) diluted in PBS at room temperature. Samples were washed 3 × 2 min in PBS then sequentially dehydrated in EtOH (50, 70, 100%) and rehydrated in hydrogen peroxide (ACD). Samples were washed twice in distilled water then PBST (0.1% Tween-20). If proteins were co-detected, then antibodies were diluted in Co-Detection Antibody Diluent (ACD; 323160). Antibodies were incubated overnight at 4 °C after the hydrogen peroxide step. Samples were washed 3 × 2 min in PBST. Then the antibodies were fixed for 30 min using 4% formaldehyde then washed 3 × 2 min in PBST. Protease IV (ACD) was applied for 30 min at RT washed 3× in deionized water. Primary RNA probes (Supplementary Table 3) were applied, and the detection protocol was completed according to manufacturer's instructions. Tissues were then counterstained with AF488 conjugated wheat germ agglutin, fluorescently labelled secondary antibodies, and DAPI.

PMOs were detected using the RNAscope Plus smRNA-RNA HD detection kit (ACD; 322780) according to the protocol described above and the manufacturer's instructions. PMO detection probes were used at 1:1000 of the manufacturer's recommended concentration.

For RNA detection in NIH/3T3 cells or mouse keratinocytes, cells were plated on Collagen IV-coated (10 μg/mL; Sigma; C5533) #1.5 glass coverslips for 2–3 h at 37 °C. Cells were fixed for 20 min at RT in 4% paraformaldehyde (EMS; 15710) then processed using the ViewRNA ISH Cell assay kit (Thermo Fisher Scientific; QVC0001) according to the manufacturer's instructions and probes for individual mRNAs (Supplementary Table 3). Cell mask and DAPI were used as counterstain to detect the cell periphery and nucleus. Samples were mounted in prolong gold.

## RNA isolation and ddPCR

Bulk RNA was isolated from in vitro cultured cells using either Trizol LS (Thermo Fisher Scientific; 10296010) or RNeasy Plus Mini kit (Qiagen; 74134). 1 μg of RNA was reverse transcribed using the iScript cDNA Synthesis Kit (Bio-Rad; 1768891) and used for droplet digital polymerase chain reaction (ddPCR). PCR reactions were prepared with cDNA, gene specific primers, and ddPCR EvaGreen Supermix (Bio-Rad; 186-4034). Droplets were generated from this reaction using the Automated Droplet Generator (Bio-Rad; 186-4101), PCR was then performed on a C1000 Touch Thermal Cycler (Bio-Rad; 185-1197), and droplet reading was performed on QX-200 Droplet Reader (Bio-Rad; 186-4003). Results were quantified using the QuantaSoft software (Bio-Rad).

## Protrusion/cell body isolation

Protrusion and cell bodies were isolated from serum-starved NIH/3T3 cells plated on transwell inserts with 3.0 μm porous polycarabonate membrane (Corning) as previously described[12]. Extracts were isolated using Trizol LS. Isolated RNAs were then quantified using the nanoString nCounter analysis platform and a custom-made codeset, according to the manufacturer's instructions.

## Western blot

For western blotting the following primary antibodies were used: rabbit anti-NET1(1:1000; Bethyl; A303-138A) and mouse anti-α-tubulin (1:2000; Sigma; T6199). Anti-rabbit and anti-mouse secondary antibodies from Li-Cor were used at 1:10,000. Membranes were scanned using an Odyssey fluorescent scanner (Li-Cor) and bands were quantified using ImageStudioLite (Li-Cor).

## Human and primate tissues

De-identified human samples were collected from normal tissue following Mohs surgery for skin cancer removal. Frozen skin samples were collected and frozen in OCT before sectioning and confirmed to be normal by H&E staining. Under National Institutes of Health protocols, the use of biospecimens from de-identified discarded human tissue does not meet the regulatory criteria for human subject research and therefore institutional review board review or informed consent are waived.

Non-human primate tissues were obtained from the National Institute on Aging's (NIA) Nonhuman Primate Tissue Bank. Fresh frozen tongue biopsies were obtained from 14 to 24 year-old female and male *Papio Anubis* (Olive Baboons) in the process of necropsy.

## Statistics and reproducibility

Where relevant, data are shown as mean ± standard error of the mean (SEM). Individual measurements are shown with differing colors corresponding to individual experiments or animals. Source data are included in the supplement. For statistical comparisons parametric tests were used to evaluate significant differences in normal distributed, homoscedastic datasets. Where either of these assumptions were not true, we instead performed non-parametric tests suitable for the data distribution. Specific statistical tests for each experiment are described in the figure captions. Animal experiments were performed in both male and female mice, but no sex-specific effects were detected so only the aggregated data are shown. In cases where representative images are shown (Figs. 1a, b and h, 2a–d, 3b, 4a, 5b, 5g) similar results were observed in multiple ROIs from samples from at least 3 mice.

## Reporting summary

Further information on research design is available in the Nature Portfolio Reporting Summary linked to this article.

# Data availability

Previously published sequencing data that were reanalyzed here are available under accession codes GSE129218 and GSE77197 in the NCBI Gene Expression Omnibus (GEO) database. All other data supporting the findings of this study are contained within the paper and its supplementary files. Source data are provided with this paper.

# Code availability

All custom scripts are provided as supplementary files.

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

## Acknowledgements

We thank the CCR Genomics Core of the National Cancer Institute, NIH for ddPCR and nanoString nCounter analysis, and the microscopy facility of the Laboratory of Cancer Biology and Genetics for Axioscan imaging. This research was made possible in part using biomaterials from the NIA Nonhuman Primate Tissue Bank (https://www.nia.nih.gov/research/dab/nonhuman-primate-tissue-bank) at the Wisconsin National Primate Research Center, University of Wisconsin-Madison under contractual agreement with the National Institute on Aging (NIA). This work was funded by the Intramural Research Program of the Center for Cancer Research, National Cancer Institute (NCI), National Institutes of Health (NIH) (1ZIA BC011501 to S.M., ZIA BC 011763 to R.B., ZIA BC 011459 to J.L. and ZIA BC 011682 to R.W.), and by R01-AR067203 and R01-AR081081 to T.L.

## Author contributions

D.E.M. and S.M. conceived the project and designed experiments. D.E.M., T.D.M., D.K., and J.L. performed experiments. D.E.M., A.N.G., N.J., T.L., R.W., R.I-B., and S.M. analyzed data. S.M. and D.E.M. wrote original manuscript. All authors reviewed and edited the manuscript.

## Funding

## Competing interests

The authors declare no competing interests.
