## [Peer Review file · Nature Communications]

Control of Epithelial Tissue Organization by mRNA Localization

Corresponding Author: Dr Stavroula Mili

Version 0:

Reviewer comments:

Reviewer #1

(Remarks to the Author)

In their manuscript, Mason et al. make the interesting observation that proper subcellular localization of Net1 mRNA is crucial for mammalian epithelial tissue organization. Many insights about the role of mRNA localization and local translation have previously been gained from studies in the brain and from experiments using in vitro systems. Here the authors focus on the role of localized mRNAs in the context of mammalian epithelial tissue homeostasis.

Overall, the manuscript is very well written, and the figures are of high quality. The experiments are performed with rigor and the results supported by proper quantifications. There are however a few critical controls that should be provided and a few suggestions and comments that should be addressed:

Figure 1:

- A negative control probe for RNA scope should be included. Maybe DapB? The authors may want to use it on all tissues shown in Fig. 1a.
- It's hard to see the basal enrichment of Net1 mRNA in the mouse skin, kidney and human skin. May want to replace by another image that is more representative of what is shown in the graph in "d".

Figure 2:

If the others want to make the point that Net1 mRNA localized to protrusions is also translated in these protrusions it would be important to show that ribosomes are present in these structures.

Extended Figure 3b, 4a:

- Net1 mRNA seems to be very low abundant in both cell lines. Please include a negative control as mentioned for Fig1a. With such a low signal it is hard to assess an enrichment in a specific area.
- Net1 mRNA is much less abundant than Cyb5r3 mRNA. Please verify by qRT-PCR
- In Extended Fig. 4a there is Net1 mRNA signal outside the cell. Is this coming from neighboring cells that were not outlined or is this background signal? Please provide an image of an area where there are no cells so that the background of the probe can be properly assessed.

Figure 4:

- Label of the images in 4b is missing

Figure 3+4:

• In Figure 3 Net1 mRNA shows basal accumulation and the protein in Figure 4 shows a cortical distribution. Line 268 states that mRNA location controls protein distribution. If so, would the protein not be expected to be more basally enriched?

Main text line 296:

Please explain what is meant by "Remarkably, this short-term induction of RhoA signaling was sufficient to significantly increase the topographical variation of the basal plasma membrane at the DEJ"

Main text line 334:

The authors write: "Targeting of the Net1 mRNA at these DEJ structures is necessary for their formation..."

I don't see any higher magnification in Fig3 that would show the loss of these protrusions. Or is this what is quantified in "k" DEJ variation?

Main text line 446:

"Phosphorylated-myosin light chain 2 (Thr18/Ser19) was detected in tissue samples using a modified immunofluorescence protocol". Are the subsequent steps the description for the sections or is this the whole tissue? It seems as if a 3 hour fixation in 4% formaldehyde is very long for a section. Please clarify.

Reviewer #2

(Remarks to the Author)

Mason et al.

Localization of mRNAs to specific subcellular regions allows synthesis of the encoded proteins to be directed to the subcellular locations where they are needed. Although this principle is well established to operate for many mRNAs, there are still not many examples of specific physiological functions that rely on this mechanism. Here, the authors study localization of the mRNA for the RhoA-GEF Net1, which they previously found to be involved in mesenchymal cell migration. They examine its role in an interesting and important system: the organization of the dermal-epidermal junction (DEJ). They report that the mRNA for Net1 localizes preferentially to basal epithelial cell protrusions that interact with the basement membrane. Experiments designed to perturb the localization of Net1 mRNA, or to alter RhoA activity, alter the structure of the DEJ.

The topic and conclusions of the study are potentially interesting. However, there is a fundamental issue that calls into question the underlying premise and logic of the study.

Major concerns

(1) Most importantly, there is a fundamental discordance between the subcellular localization of Net1 mRNA versus protein. The underlying premise of the topic is that specific mRNAs are localized subcellularly, and this results in synthesis of the encoded proteins preferentially at the same subcellular locations. As stated in the Abstract "Net1 mRNA accumulates at DEJ protrusion-like structures that interact with the basement membrane". Accordingly, Figs. 1-3 provide evidence that the Net1 mRNA is distributed along an apical-basal axis, being concentrated in basal cell layers and subcellularly in basal protrusions. However, this does not match up with the distribution analyzed for the protein, where the subcellular distribution is instead analyzed between the cell cortex versus non-cortical locations, with Net1 protein being distributed "throughout the basal keratinocyte cortex" with no asymmetry between apical and basal sides of the cell (line 265 and following; Fig. 4a-d). How do the authors imagine that an apical-basal asymmetry in Net1 mRNA localization leads to a cortical/non-cortical asymmetry in Net1 protein localization? This is entirely unexplained and would require a clear and believable model, as well as convincing experimental evidence, to show how the mRNA localization leads to the protein localization. If that's not the significance of the mRNA localization, then what is it doing? This is a crucial element for the logic of the overall story to make sense, and to be confident that the results of the individual perturbation experiments are being interpreted correctly.

(2) Fig. 1, Extended Data Fig. 1, Fig. 3a-c. From the first sentence of the Abstract onwards, the paper is framed as being about mRNA localization at a subcellular level. However, the localization analyses in these Figures don't distinguish between subcellular localization versus localization to different cells (such as the layer of cells closest to the basement membrane). This distinction needs to be made clear in the text, and at least some of the quantitative analyses in each Figure should provide a measure of specifically subcellular localization.

Other comments

- The structure of the DEJ has been examined by many light and electron microscope studies over a period of decades, as briefly mentioned in the results section (lines 153-154). Since a major claim of the study is to have identified a new kind of structure, "uncharacterized protrusion-like structures" (line 73), please comment in much more detail how the structures observed here relate to what has been seen in previous studies. Were previous studies consistent with the results here? If a major structural feature was previously missed, how did that happen? Were the structures previously visible, but just not named? Also, to what degree are the structures observed here specific to the tongue epithelium - are they also found in

skin?

- The basement membrane plays a key role in cancer. The relevance to cancer is not mentioned in the Introduction or Discussion, which could help to increase the interest level of the paper.

Version 1:

Reviewer comments:

Reviewer #1

(Remarks to the Author)

The authors have addressed all my concerns and comments.

Reviewer #2

(Remarks to the Author)

The authors have generally responded well to the comments in my initial review.

I have just one minor follow-up comment. In my initial review I had interpreted the word "uncharacterized" to mean that the authors were claiming the structures had not been observed previously. The authors respond that this had not been their intention. However, "uncharacterized" still seems inaccurate to me, since there was previously some characterization. Since other readers may interpret the word the same way that I did, I think it would be better to use a different word or phrasing such as "poorly characterized" or "not previously characterized at a molecular level" or something similar.

We would like to thank very much the Reviewers for the feedback on our work and their constructive comments. We have revised our manuscript by adding new data as well as textual clarifications, which we believe have addressed the points raised and have strengthened our conclusions. These changes are detailed in our point-by-point responses below and are highlighted in the revised text.

REVIEWER COMMENTS

Reviewer #1 (Remarks to the Author):

In their manuscript, Mason et al. make the interesting observation that proper subcellular localization of Net1 mRNA is crucial for mammalian epithelial tissue organization. Many insights about the role of mRNA localization and local translation have previously been gained from studies in the brain and from experiments using in vitro systems. Here the authors focus on the role of localized mRNAs in the context of mammalian epithelial tissue homeostasis. Overall, the manuscript is very well written, and the figures are of high quality. The experiments are performed with rigor and the results supported by proper quantifications. There are however a few critical controls that should be provided and a few suggestions and comments that should be addressed:

Figure 1:

- A negative control probe for RNA scope should be included. Maybe DapB? The authors may want to use it on all tissues shown in Fig. 1a.*

Thank you for pointing out this omission. We have included representative images of each tissue stained with negative control probes targeting the bacterial gene *DapB* in comparison to staining with *Net1/Gapdh* probes. We observe minimal, if any, background with the negative control probes, supporting the specificity of the observed signals. These data are shown in new Supplemental Figure 1 and are referenced in lines 93-95.

- It's hard to see the basal enrichment of Net1 mRNA in the mouse skin, kidney and human skin. May want to replace by another image that is more representative of what is shown in the graph in "d".*

We thank the Reviewer for their suggestion and have replaced the representative images of the mouse and human skin with regions where the basal accumulation of the Net1 mRNA is visually easier to appreciate. See these modifications in Figure 1a

With regard to the mouse kidney, we have not replaced the image, since the *Net1* mRNA distribution shown is what we consistently observe in kidney tissue. We think that, in this case, the difficulty in appreciating the basal enrichment might stem from the relative lack of clustered basal *Net1* mRNA signal as is typically seen in other tissues. Nevertheless, the more dispersed *Net1* signal seen in the kidney is still mostly observed below the nucleus towards the basal surface leading to the quantitative results shown in the graph of Figure 1d. We are intrigued by what the dispersed versus clustered mRNA distribution could signify but cannot provide further insights into that yet.

Figure 2:

If the others want to make the point that Net1 mRNA localized to protrusions is also translated in these protrusions it would be important to show that ribosomes are present in these structures.

We thank the Reviewer for the helpful suggestion. We have now demonstrated the presence of ribosomes in epithelial protrusion-like structures in two ways: a) by transmission electron microscopy (TEM) and b) by immunostaining for ribosomal proteins of the small (Rps27) and large (Rpl23a) ribosomal subunits followed by super-resolution Stimulated Emission Depletion (STED) microscopy. The rationale of this approach is that proximity of small and large subunits would indicate the presence of translating ribosomes. Indeed, both methods support the presence of ribosomes at protrusion-like structures. TEM imaging (shown in new Figure 2d) revealed that within protrusions there are several ribosomes observed as electron dense particles with an average diameter of 24.31 \pm 3.97 nm. In some instances, strings of ribosomes can be observed, likely reflecting actively translating polysomes. Additionally, STED imaging (shown in new supplemental figure S4b) revealed the presence of putative 80S complexes near the plasma membrane in protrusion-like structures, as evidenced by proximity of Rpl23a and Rps27 signals within a radius of 100nm or less, a physical distance consistent with labeling of the two proteins in the context of a fully formed 80S ribosome. These complementary methods thus support the presence of ribosomes at basal keratinocyte protrusions. This is consistent with the model that protrusion-like structures can support local translation of mRNAs like *Net1*. These results are described in lines 200-213.

Extended Figure 3b, 4a:

• Net1 mRNA seems to be very low abundant in both cell lines. Please include a negative control as mentioned for Fig1a. With such a low signal it is hard to assess an enrichment in a specific area.

To address this concern, we have included additional controls. We have used two different siRNAs targeting *Net1*. Knockdown of *Net1* mRNA using these siRNAs effectively reduced *Net1* mRNA signal without changing *Cyb5r3* mRNA in 3T3 fibroblasts, demonstrating the specificity of the *Net1* mRNA probes used for RNA detection. These data are shown in new Supplemental Figure 5a-c and are referenced in lines 225-226.

• Net1 mRNA is much less abundant than Cyb5r3 mRNA. Please verify by qRT-PCR

Indeed, *Cyb5r3* is ~20 times more abundant than *Net1* in NIH/3T3 fibroblasts, consistent with the FISH imaging data. We have assessed this using direct mRNA counts from nanoString nCounter assays. These data are shown in new Supplemental Figure 5d.

• In Extended Fig. 4a there is Net1 mRNA signal outside the cell. Is this coming from neighboring

cells that were not outlined or is this background signal? Please provide an image of an area where there are no cells so that the background of the probe can be properly assessed.

The Reviewer is correct that there are additional cells in two of the images, as shown below. The figure in the manuscript highlights in each image only the cell that was being analyzed. Overall, the FISH signals we detect are largely specific (as supported by the siRNA experiments described above) and there is minimal background signal outside of cell borders. The single spot outside of the cell boundary seen in the right panel below, is a rare event. We have opted to keep the existing figures in place (supplemental figure 6a).

Figure 4:

• Label of the images in 4b is missing

Thank you for the correction to figure 4b, this has been addressed.

Figure 3+4:

• In Figure 3 Net1 mRNA shows basal accumulation and the protein in Figure 4 shows a cortical distribution. Line 268 states that mRNA location controls protein distribution. If so, would the protein not be expected to be more basally enriched?

The Reviewer is correct that, in some instances, mRNA translation at a specific subcellular location leads to accumulation of the encoded protein at the same site. Nevertheless, it has also become clear in recent years that this is not always the case. Multiple genome-wide studies have shown that local mRNA accumulation often does not correlate with local protein abundance (e.g. Mardakheh et al. Dev Cell 2015, Moor et al. Science 2017, Chouaib et al. Dev Cell 2020, Novoselsky et al Plos Bio. 2024). This discordance occurs even though the locally targeted mRNA is actively being translated. Several studies, focusing on individual mRNAs, have provided an explanation for how in these discordant cases local mRNA translation can influence protein function (Moissoglu et al. EMBO 2020, Gasparski et al. Mol Cell 2023, Norris et al. Genes Dev. 2023). The common theme emerging from these studies is that local protein synthesis is important not for directing the local accumulation

of a protein at the same site but rather for kinetically favoring the co-translational interaction of the newly-synthesized protein with specific partners. The protein can then diffuse away from the translation site and adopt a distribution pattern distinct from that of its mRNA. Importantly, if mRNA translation occurs in a different subcellular location, then association with partners changes. In this way, local synthesis can affect protein function even if the protein is not retained close to the site of translation.

In the case of Net1, the differential association of the protein with binding partners based on mRNA location, leads to a differential distribution of Net1 protein between the cytoplasm/cell cortex and the nucleus. Specifically, we had previously shown (Gasparski et al. Mol Cell 2023) that in *in vitro* cultured mesenchymal cells the *Net1* mRNA accumulates and is translated at peripheral protrusions (without leading to an accumulation of Net1 protein at these sites). Instead, *Net1* mRNA translation at peripheral protrusions is important because it favors interaction of the Net1 protein with a membrane-bound scaffold, CASK, thus retaining Net1 protein in the cytoplasm where it functions to activate RhoA and promote cell migration. We emphasize again that Net1 protein retained in the cytoplasm does not exhibit any observable local accumulation, likely due to diffusion away from translation sites. If, on the other hand, *Net1* mRNA distribution is altered so that it is more perinuclear, then the newly-synthesized Net1 protein preferentially interacts with different partners, members of the importin transport complex, resulting in Net1 import and sequestration into the nucleus and thus reduced RhoA activity.

As we show here, in keratinocytes of epithelial tissues, Net1 is synthesized near basal protrusion-like structures and adopts a cytoplasmic/cortical localization. Mislocalization of the *Net1* mRNA away from these basal areas leads to reduced cortical and increased nuclear distribution of the protein and is accompanied by corresponding changes in RhoA signaling. We believe that the work presented in this manuscript is in full agreement with the known *in vitro* molecular understanding and current views of the field and significantly extends them by showing that this mRNA-location-based regulation is not solely an *in vitro* phenomenon but occurs in physiological tissues and has consequences for tissue physiology and function. We have tried to better explain these concepts in the revised text (lines 59-71 and 299-303).

The same point was brought up by Reviewer #2 (major concern 1).

Main text line 296:

Please explain what is meant by “Remarkably, this short-term induction of RhoA signaling was sufficient to significantly increase the topographical variation of the basal plasma membrane at the DEJ”

To clarify, we find remarkable that doxycycline induction of ArhGEF11, a RhoA activator, for as little as 10 hours causes DEJ remodeling. We had not expected such a quick and obvious change *in vivo*, where keratinocytes are firmly adhered to the basement membrane. This

effect was most obvious as an increase in the degree to which the basal plasma membrane of keratinocytes extends into the stroma. Quantitatively we measure this using a metric we call 'DEJ variation' which provides an approximate measurement of 2D topography of the basal plasma membrane. We use the term topography analogously to how the term is used in material science to characterize the features of a surface. A schematized explanation of how the 'DEJ variation' metric is derived is provided in Supplementary Figure S7. Briefly, 'DEJ variation' expresses the distances in peaks and valleys between plasma membrane protrusions. It is equivalent to the 'Average roughness (Ra)', a common measurement of surface texture.

We have modified the text to clarify these points in lines 261- 265 and 334-335 and have added further details in the Methods about the DEJ variation analysis (lines 577-587).

Main text line 334:

The authors write: "Targeting of the Net1 mRNA at these DEJ structures is necessary for their formation..."

I don't see any higher magnification in Fig3 that would show the loss of these protrusions. Or is this what is quantified in "k" DEJ variation?

The Reviewer is correct. We use the metric of 'DEJ variation' to quantitatively assess differences in the topography of the basal plasma membrane. As detailed above, the values of 'DEJ variation' provide an indicator of the degree to which basal keratinocytes are producing protrusion-like structures and effectively "shaping" the DEJ.

To additionally facilitate the visual assessment of the differences in DEJ structure we have provided high-resolution serial optical slices of the DEJ for each of the conditions shown in Figure 3j. These are shown in the new Supplemental video 3 and referenced in lines 261-262. We hope they better highlight the changes in plasma membrane topography upon delivery of Net1 PMOs.

Main text line 446:

"Phosphorylated-myosin light chain 2 (Thr18/Ser19) was detected in tissue samples using a modified immunofluorescence protocol". Are the subsequent steps the description for the sections or is this the whole tissue? It seems as if a 3 hour fixation in 4% formaldehyde is very long for a section. Please clarify.

During initial optimization of the pMLC staining protocol we found that tissue sections fixed for only 20-30 minutes were damaged by subsequent treatment with 1% SDS, which is used for antigen retrieval. We thus opted to extend tissue fixation to 3 hours. This extended fixation prevented tissue destruction during antigen retrieval and showed excellent and specific staining with the pMLC antibody (as shown in Supplemental Figure S9). Shorter duration fixation, without antigen retrieval, improved staining quality but not to the same extent as extended fixation followed by SDS. We note that we have not determined the minimum duration of fixation required for high-quality staining, but 3 hrs was used for the data collected in Figures 4, 5 and Supplemental Figure S9.

Reviewer #2 (Remarks to the Author):

Mason et al.

Localization of mRNAs to specific subcellular regions allows synthesis of the encoded proteins to be directed to the subcellular locations where they are needed. Although this principle is well established to operate for many mRNAs, there are still not many examples of specific physiological functions that rely on this mechanism. Here, the authors study localization of the mRNA for the RhoA-GEF Net1, which they previously found to be involved in mesenchymal cell migration. They examine its role in an interesting and important system: the organization of the dermal-epidermal junction (DEJ). They report that the mRNA for Net1 localizes preferentially to basal epithelial cell protrusions that interact with the basement membrane. Experiments designed to perturb the localization of Net1 mRNA, or to alter RhoA activity, alter the structure of the DEJ.

The topic and conclusions of the study are potentially interesting. However, there is a fundamental issue that calls into question the underlying premise and logic of the study.

Major concerns

(1) Most importantly, there is a fundamental discordance between the subcellular localization of Net1 mRNA versus protein. The underlying premise of the topic is that specific mRNAs are localized subcellularly, and this results in synthesis of the encoded proteins preferentially at the same subcellular locations. As stated in the Abstract “Net1 mRNA accumulates at DEJ protrusion-like structures that interact with the basement membrane”. Accordingly, Figs. 1-3 provide evidence that the Net1 mRNA is distributed along an apical-basal axis, being concentrated in basal cell layers and subcellularly in basal protrusions. However, this does not match up with the distribution analyzed for the protein, where the subcellular distribution is instead analyzed between the cell cortex versus non-cortical locations, with Net1 protein being distributed “throughout the basal keratinocyte cortex” with no asymmetry between apical and basal sides of the cell (line 265 and following; Fig. 4a-d). How do the authors imagine that an apical-basal asymmetry in Net1 mRNA localization leads to a cortical/non-cortical asymmetry in Net1 protein localization? This is entirely unexplained and would require a clear and believable model, as well as convincing experimental evidence, to show how the mRNA localization leads to the protein localization. If that’s not the significance of the mRNA localization, then what is it doing? This is a crucial element for the logic of the overall story to make sense, and to be confident that the results of the individual perturbation experiments are being interpreted correctly.

The Reviewer is correct that, in some instances, mRNA translation at a specific subcellular location leads to accumulation of the encoded protein at the same site. Nevertheless, it has also become clear in recent years that this is not always the case. Multiple genome-wide studies have shown that local mRNA accumulation often does not correlate with local protein abundance (e.g. Mardakheh et al. Dev Cell 2015, Moor et al. Science 2017, Chouaib et al. Dev Cell 2020, Novoselsky et al Plos Bio. 2024). This discordance occurs even though the locally targeted mRNA is actively being translated. Several studies, focusing on individual mRNAs, have provided an explanation for how in these discordant cases local mRNA translation can influence protein function (Moissoglu et al. EMBO 2020, Gasparski et

al. Mol Cell 2023, Norris et al. Genes Dev. 2023). The common theme emerging from these studies is that local protein synthesis is important not for directing the local accumulation of a protein at the same site but rather for kinetically favoring the co-translational interaction of the newly-synthesized protein with specific partners. The protein can then diffuse away from the translation site and adopt a distribution pattern distinct from that of its mRNA. Importantly, if mRNA translation occurs in a different subcellular location, then association with partners changes. In this way, local synthesis can affect protein function even if the protein is not retained close to the site of translation.

In the case of Net1, the differential association of the protein with binding partners based on mRNA location, leads to a differential distribution of Net1 protein between the cytoplasm/cell cortex and the nucleus. Specifically, we had previously shown (Gasparski et al. Mol Cell 2023) that in *in vitro* cultured mesenchymal cells the *Net1* mRNA accumulates and is translated at peripheral protrusions (without leading to an accumulation of Net1 protein at these sites). Instead, *Net1* mRNA translation at peripheral protrusions is important because it favors interaction of the Net1 protein with a membrane-bound scaffold, CASK, thus retaining Net1 protein in the cytoplasm where it functions to activate RhoA and promote cell migration. We emphasize again that Net1 protein retained in the cytoplasm does not exhibit any observable local accumulation, likely due to diffusion away from translation sites. If, on the other hand, *Net1* mRNA distribution is altered so that it is more perinuclear, then the newly-synthesized Net1 protein preferentially interacts with different partners, members of the importin transport complex, resulting in Net1 import and sequestration into the nucleus and thus reduced RhoA activity.

As we show here, in keratinocytes of epithelial tissues, Net1 is synthesized near basal protrusion-like structures and adopts a cytoplasmic/cortical localization. Mislocalization of the *Net1* mRNA away from these basal areas leads to reduced cortical and increased nuclear distribution of the protein and is accompanied by corresponding changes in RhoA signaling. We believe that the work presented in this manuscript is in full agreement with the known *in vitro* molecular understanding and current views of the field and significantly extends them by showing that this mRNA-location-based regulation is not solely an *in vitro* phenomenon but occurs in physiological tissues and has consequences for tissue physiology and function. We have tried to better explain these concepts in the revised text (lines 59-71 and 299-303).

(2) Fig. 1, Extended Data Fig. 1, Fig. 3a-c. From the first sentence of the Abstract onwards, the paper is framed as being about mRNA localization at a subcellular level. However, the localization analyses in these Figures don't distinguish between subcellular localization versus localization to different cells (such as the layer of cells closest to the basement membrane). This distinction needs to be made clear in the text, and at least some of the quantitative analyses in each Figure should provide a measure of specifically subcellular localization.

We believe there might be some misunderstanding regarding the quantitative analyses presented in our work. Figures 1d, e, g, Supplemental Figure s2c and Figures 3c, d all indeed present a measure of **subcellular** mRNA localization. We think that our analysis process might have not come across very clearly. The process consists of a) manually segmenting the layer of cells that are in contact with the basement membrane, b) within this layer (which is one-cell high) we measure the RNA amount across the apico-basal axis and, c) we define the RNA fraction in the bottom 30% of this cell layer as the 'basal RNA fraction'. This process thus provides a measure of subcellular mRNA distribution (not within a single cell but across a layer of similarly polarized individual cells). We have made changes in the text to clarify how the analysis is done and what exactly is being measured (lines 100-105). We have also added in Figure 1 a schematic (new panel c) depicting how the subcellular localization of mRNA was measured to yield the "basal mRNA fraction" metric. We have also changed the axes label in Supplemental Figure 2e, which is the only instance of total RNA measurement across a cell layer, to avoid confusion between this total intensity measurement and the basal mRNA fraction used in all other graphs.

Other comments

- The structure of the DEJ has been examined by many light and electron microscope studies over a period of decades, as briefly mentioned in the results section (lines 153-154). Since a major claim of the study is to have identified a new kind of structure, "uncharacterized protrusion-like structures" (line 73), please comment in much more detail how the structures observed here relate to what has been seen in previous studies. Were previous studies consistent with the results here? If a major structural feature was previously missed, how did that happen? Were the structures previously visible, but just not named? Also, to what degree are the structures observed here specific to the tongue epithelium - are they also found in skin?

It was not our intention to suggest that we have identified a new kind of structure. The Reviewer is correct that these structures have been previously observed by electron microscopy. See for example images below from older or more recent studies - that we cite in the text - and the various ways in which the authors had described these formations:

[Redacted]

These studies had pointed out that how pronounced these protrusion-like structures are depends on the anatomical location in the skin. However, we were unable to find any other information on the composition or regulation of these structures, potentially because they are best visualized by electron microscopy or high-resolution confocal microscopy techniques which are not typically used in current studies of epithelial biology. We had thus referred to them as ‘uncharacterized protrusion-like structures’ to highlight the lack of information about them. We believe that this is an accurate way of representing the available knowledge thus far and have made slight clarifications to the text. We also think that our demonstration that these structures serve as areas for mRNA deposition/translation and that they are modulated by *Net1* mRNA targeting significantly advances our understanding of the regulation and potential roles of these formations.

Regarding the appearance of these structures in other epithelia, we agree with the Reviewer that this is an important question. We have stained tail skin tissue for integrin $\alpha 6$, WGA and *Net1* mRNA, and have provided high resolution confocal images (new Supplemental Figure 3; referenced in lines 177-183). We observe similar protrusion-like structures of the basal plasma membrane and accumulation of *Net1* mRNA in areas where WGA-positive fibers from the dermis make contact with keratinocytes. These data suggest that our observations are broadly observed in stratified epithelial tissues.

- The basement membrane plays a key role in cancer. The relevance to cancer is not mentioned in the Introduction or Discussion, which could help to increase the interest level of the paper.

We appreciate the reviewer’s suggestion on expanding our discussion to include the possible relevance of our findings to cancer. We have added additional commentary in the discussion in lines 387-393 to address this.